# The Role of Viral Infections in the Immunopathogenesis of Type 1 Diabetes Mellitus: A Narrative Review

**DOI:** 10.3390/biology14080981

**Published:** 2025-08-02

**Authors:** Ioanna Kotsiri, Maria Xanthi, Charalampia-Melangeli Domazinaki, Emmanouil Magiorkinis

**Affiliations:** 1Second Department of Internal Medicine, Asklepieion General Hospital Voulas, 16673 Athens, Greece; ikotsiri@yahoo.com (I.K.); hara_domaz@yahoo.gr (C.-M.D.); 2Second Department of Internal Medicine, General Hospital Tzaneio, 18537 Pireas, Greece; xanthimary@gmail.com; 3Department of Laboratory Haematology, Metaxas Anticancer Memorial Hospital, 18537 Pireas, Greece

**Keywords:** type 1 diabetes mellitus, viral infections, enteroviruses, Coxsackie B virus, autoimmunity, beta-cell destruction, molecular mimicry, SARS-CoV-2, HLA genotype, environmental triggers

## Abstract

Type 1 diabetes is a lifelong condition that begins when the body’s immune system mistakenly destroys insulin-producing beta cells in the pancreas. This review examines the potential role of common viral infections in triggering or accelerating this autoimmune response. Key findings include that enteroviruses, especially Coxsackie B viruses, show the strongest association with new-onset type 1 diabetes. Other viruses—including mumps, rubella, rotavirus, influenza, and SARS-CoV-2—have also been linked, though less consistently. Interestingly, some infections, such as cytomegalovirus or chickenpox, may even delay disease onset. Viruses may influence disease development by directly damaging pancreatic beta cells or by triggering immune confusion through molecular mimicry, where viral proteins resemble human ones. While no single virus is definitively proven to cause the disease, infections appear to accelerate or precipitate diabetes in genetically at-risk individuals. Understanding these interactions could support vaccine development, antiviral therapies, and early screening programs. Such approaches may help reduce the incidence or delay the onset of type 1 diabetes and its burden on individuals and healthcare systems.

## 1. Introduction

Type 1 diabetes mellitus (T1DM) is a chronic autoimmune disease characterized by the selective destruction of insulin-producing beta cells in the pancreatic islets of Langerhans, leading to absolute insulin deficiency. T1DM accounts for approximately 5–10% of all diabetes cases and typically presents in children and young adults under the age of 40 [1]. Clinical onset occurs after the loss of 80–90% of beta-cell mass, resulting in hyperglycemia and the need for lifelong insulin therapy [1].

T1DM is widely recognized as a multifactorial disease resulting from the interplay between genetic susceptibility and environmental triggers as initiators or accelerators of the autoimmune process. Genetic studies have identified a strong association with the human leukocyte antigen (HLA) system, particularly class II alleles (e.g., HLA-DR and HLA-DQ), which are linked to an increased risk of autoimmune responses against pancreatic islet cells [2].

The autoimmune process underlying T1DM is gradual and often begins years before the appearance of clinical symptoms. During this latent phase, the immune system produces autoantibodies directed against islet cell antigens, such as insulin, glutamic acid decarboxylase (GAD), and tyrosine phosphatase (IA-2). Cellular immunity also plays a central role, involving the activation of T lymphocytes and inflammatory cytokines that progressively damage beta cells [3,4].

Although genetic susceptibility, particularly involving HLA class II alleles, is a well-established risk factor, it is insufficient on its own to cause disease [5,6]. The onset of T1DM clearly requires additional environmental triggers. Among these, viral infections have emerged as key suspects, supported by decades of epidemiological and experimental research [7,8,9]. Seasonal patterns in T1DM incidence, as well as geographic variations, further support the hypothesis that viral agents may serve as key environmental triggers, especially in genetically predisposed individuals [1,4].

In light of the COVID-19 pandemic [10,11] and recent advances in molecular diagnostics [1,12], a comprehensive synthesis of current knowledge is both timely and necessary.

This narrative review aims to achieve the following:Summarize the major viral pathogens implicated in T1DM;Explain proposed immunopathogenic mechanisms;Highlight key population studies and recent evidence.

While this is not a systematic review, we applied a structured literature search strategy using PubMed, Scopus, and Google Scholar for English-language studies published between 1990 and 2024. Search terms included the following: “Type 1 Diabetes Mellitus” AND (“viral infection” OR “enterovirus” OR “Coxsackievirus” OR “molecular mimicry” OR “autoimmunity” OR “SARS-CoV-2” OR “rotavirus” OR “mumps” OR “rubella” OR “cytomegalovirus” OR “influenza”).

We prioritized peer-reviewed cohort studies, experimental models, and mechanistic investigations relevant to beta-cell autoimmunity. Case reports, non-peer-reviewed preprints, and studies lacking direct relevance to T1DM pathogenesis were excluded.

## 2. Pathogenesis of T1DM

T1DM is an autoimmune disease in which abnormal activation of both the innate and adaptive immune systems leads to the destruction of pancreatic beta cells. Key players include autoreactive T and B lymphocytes, natural killer (NK) cells, and antigen-presenting cells such as dendritic cells. NK cells serve as a critical bridge between innate and adaptive immunity and appear early in the islets during the preclinical phase of disease. Their cytotoxic activity, along with their interactions with dendritic cells and T cells, contributes to the development of islet-directed autoimmunity [5].

Genetic predisposition—particularly HLA genotypes—primes individuals for autoimmunity, but environmental triggers such as viral infections are often required to initiate the immune cascade. These infections activate a broad range of immune components, including Th1 and Th2 cells, CD8+ cytotoxic T cells, NK cells, macrophages, and proinflammatory cytokines, such as IL-1, IL-6, and TNF-α. Th1 lymphocytes promote inflammation and cytotoxicity, while Th2 cells may exert regulatory functions. In genetically susceptible individuals, viral antigens presented via HLA class II molecules can mimic beta-cell epitopes, contributing to the activation of autoreactive T cells and the production of islet autoantibodies [2]. NK cells are consistently found in the pancreas during disease progression and contribute to beta-cell destruction both directly and indirectly through the activation of cytotoxic T lymphocytes [4,5]. Regulatory T-cell dysfunction and altered antigen presentation further exacerbate the autoimmune response, leading to progressive beta-cell loss and clinical onset of diabetes.

## 3. Viral Infections and Type 1 Diabetes (T1DM)

Viral infections are among the most widely studied environmental triggers of T1DM. Variability in disease incidence across regions and seasons has long supported a potential viral contribution [1]. A range of viruses—including enteroviruses (especially Coxsackie B4), mumps, rubella, cytomegalovirus (CMV), rotavirus, influenza, and SARS-CoV-2—have been implicated in disease onset [1].

Rather than examining each virus in isolation, this review organizes findings based on the following three major immunopathogenic mechanisms:Direct cytolysis: viral infection of pancreatic beta cells leading to functional impairment or lysis;Molecular mimicry: cross-reactivity between viral antigens and beta-cell autoantigens;Persistent infection and immune modulation: chronic low-grade viral presence that sustains immune activation or disrupts tolerance.

While the strength of association varies by virus, studies such as DiViD [1,13], TEDDY [2], MIDIA [14], and DAISY [4] suggest that viral infections—especially by enteroviruses—may act as initiators or accelerators of autoimmunity in genetically predisposed individuals. However, establishing causality remains challenging due to variations in study design, detection methods, and population heterogeneity [15].

Table 1 summarizes the current knowledge about the role of viral infections in the pathogenesis of T1DM.

### 3.1. Direct Cytolysis: Enteroviruses—Coxsackie-B (CVBs), Mumps Virus, SARS-CoV-2

Enteroviruses are widespread, non-enveloped, and single-stranded RNA viruses that are primarily transmitted via the fecal-oral route and replicate mainly in the gastrointestinal tract. More than 100 different serotypes affecting humans have been identified. Although most enteroviral infections are asymptomatic, certain serotypes can cause severe clinical manifestations in a small percentage of those infected. Enteroviruses are most easily detected in stool, where they can usually be found for 3–4 weeks but rarely beyond 2–3 months. They may also be detected in blood samples [1,21].

Both in vitro and ex vivo studies have shown that various viruses can infect pancreatic beta cells, causing a range of effects from functional impairment to apoptotic cell death. Human enteroviruses—particularly Coxsackie viruses—have been shown to significantly affect both the exocrine and endocrine pancreatic cells, including beta cells, by inducing a localized inflammatory response [12].

Enteroviruses, especially Coxsackie B viruses, have been associated with the development of type 1 diabetes mellitus (T1DM). Natural killer (NK) cells appear to play a critical role in the immunopathogenesis of the disease. In patients with T1DM, a reduction in both the number and functionality of NK cells has been observed, along with altered receptor expression. Specifically, decreased expression of activating receptors (such as NKG2D, NKp46, and NKp30) and increased expression of inhibitory receptors (such as KIR2DL3) have been reported. These disturbances lead to diminished cytolytic activity of NK cells against pancreatic beta cells infected with Coxsackie B viruses, allowing persistent viral infection and triggering autoimmune responses that contribute to beta-cell destruction [5].

The DiViD study, launched in Norway, aimed to determine whether viral infections—primarily enteroviruses—are involved in the pathogenesis of type 1 diabetes. It included six adult patients (aged 20–35) newly diagnosed with T1DM within nine months of diagnosis. Pancreatic biopsies were obtained laparoscopically. The study detected enteroviral RNA, primarily Coxsackie B, in pancreatic tissue in three of the six patients. Viral RNA was mainly found in the beta cells of the islets, which are affected in T1DM. The presence of viral proteins alongside inflammatory markers suggests a possible role for the virus in disease onset. However, active viral replication was not observed, supporting the hypothesis of a persistent infection in the pancreatic tissue [1].

The TEDDY study, which focused on children genetically predisposed to T1DM, demonstrated that recent viral infections—such as those from enteroviruses and influenza viruses—may act as triggers for autoimmunity, though no single causative virus has been identified [10]. Similarly, the MIDIA study focused on the interaction between viral infections and immune responses in adolescents. The presence of viral RNA and activation of T lymphocytes against viral antigens suggest that specific viruses—particularly enteroviruses—may actively participate in initiating or accelerating the autoimmune destruction of pancreatic beta cells [2].

The DAISY (Diabetes Autoimmunity Study in the Young) study, a long-term prospective epidemiological study, aimed to determine whether enteroviral infection predicts the progression from islet autoimmunity to clinical T1DM in genetically at-risk children. A total of 2365 children with genetic susceptibility (based on HLA or family history) were included. Of these, 140 developed persistent islet autoantibodies (GAD, insulin, and IA-2). Blood and stool samples were collected every 3–6 months, and enteroviral RNA was detected using RT-PCR. Over 4.2 years, 50 children progressed to T1DM. The presence of enteroviral RNA in the blood was associated with a significantly increased risk of developing diabetes (HR = 7.02). No corresponding association was found for RNA in stool or evidence of chronic infection. This study highlighted that enteroviral infection in the blood may accelerate the onset of T1DM in genetically susceptible children with existing autoimmunity [4].

In another study, Helfand et al. [22] investigated the association between enteroviral infection and the onset of T1DM. The study included 128 patients aged ≤18 years with newly diagnosed insulin-dependent diabetes mellitus (IDDM) and 120 age-, sex-, and serum-date-matched healthy controls. Serological testing showed increased titers of enterovirus antibodies in recently diagnosed T1DM patients compared to healthy individuals. The findings support the hypothesis that enteroviruses may be involved in the onset of T1DM, possibly through immune mechanisms leading to beta-cell destruction [22].

The review by Roivainen [21] summarizes the latest findings linking enteroviral infections—particularly Coxsackie B viruses—to the pathogenesis of T1DM. It highlights that enteroviruses exhibit tropism for pancreatic beta cells and that their genetic material and viral antigens have been detected in pancreatic tissues of individuals with T1DM. Experimental models have shown that these viruses can cause functional damage or apoptosis of beta cells, contributing to their autoimmune destruction [21].

Several studies have documented the presence of CVB antigens inside pancreatic beta cells during infection, suggesting that these viruses are capable of infecting beta cells both in vivo and in vitro. This hypothesis is further supported by epidemiological observations indicating that seasonal peaks of enteroviral infections—especially CVBs—coincide with increased incidence of insulin-dependent diabetes mellitus (IDDM) [21,23,24].

The study by Cinek and colleagues [25] monitored 911 Norwegian infants with the highest genetic risk for developing T1DM. Fecal samples were repeatedly tested for enteroviruses and adenoviruses. The findings showed that enteroviruses were frequently present, while adenoviruses were less common. Although enteroviral infections were prevalent, no clear association was found between virus presence and the development of autoimmune markers or the onset of T1DM. These results suggest that enteroviral infection is common in high-risk infants, but its direct link to T1DM onset remains unclear [25].

As one can notice, whereas multiple studies such as DiViD [1] and DAISY [4] have demonstrated a strong association between enterovirus infection and the development or progression of T1DM, other studies, notably by Cinek et al. [25], have failed to confirm this link. These discrepancies likely stem from differences in study design, including sample collection timing (e.g., blood vs. stool), detection sensitivity (RT-PCR protocols), genetic risk stratification, and population heterogeneity. For example, the DAISY study detected enteroviral RNA in blood, correlating with disease progression, whereas Cinek et al. used stool samples and found no significant association with islet autoimmunity or diabetes onset. This highlights the importance of tissue-specific viral detection and study timing in understanding enterovirus–T1DM interactions.

The prospective study by Hyöty et al. [26], part of the DiMe (Childhood Diabetes in Finland) program, investigated the role of enteroviral infections—mainly Coxsackie B virus—in the pathogenesis of T1DM. It included children with T1DM, high-risk siblings, and healthy controls. The findings showed a significantly higher frequency of recent enteroviral infections in children who developed T1DM, particularly under the age of three, as well as associations with intrauterine exposure to enteroviruses. The study supports the hypothesis that enteroviruses may contribute to the initiation of the autoimmune process leading to beta-cell destruction and the development of T1DM [26].

Mumps virus outbreaks have been associated with an increase in the incidence of T1DM, with a delay of 2 to 4 years. Experimental data have shown that the mumps virus can infect human pancreatic beta cells in vitro and increase the expression of HLA class I molecules.

The study by Hyöty et al. [15] examined the relationship between antibodies to the mumps virus and the incidence of T1DM in Finland. The results showed reduced levels of mumps antibodies in children with T1DM, along with a stabilization in the rising incidence of T1DM following the introduction of the MMR (measles–mumps–rubella) vaccine. The study supports the hypothesis that the mumps virus may be involved in the development of T1DM and that vaccination may have contributed to curbing the rising incidence of the disease in Finland [15].

The study by Parkkonen et al. [16] investigated the ability of the mumps virus to infect beta cells in cultures of human fetal pancreatic islets. The results showed that the virus can directly infect beta cells and increase the expression of HLA class I molecules. This upregulation may render beta cells more vulnerable to immune system attack, contributing to the autoimmune destruction that leads to T1DM. The study suggests that the mumps virus may be involved in T1DM pathogenesis through immune mechanisms triggered by infection of beta cells [16].

Severe acute respiratory syndrome coronaviruses (SARS-CoV and SARS-CoV-2) can enter islet cells through angiotensin-converting enzyme 2 (ACE2) receptors and cause reversible damage to beta cells and transient hyperglycemia [27]. This process is accompanied by a localized inflammatory response, with increased production of proinflammatory cytokines, such as IL-6, TNF-α, and IL-1β, contributing to further metabolic dysregulation. Beta-cell damage may lead to transient hyperglycemia or, in some cases, the onset of new diabetes [28].

During the 2003 SARS outbreak caused by SARS-CoV, cases of acute hyperglycemia were observed. In a study by Yang et al. [20], it was found that during systemic infection, the virus binds to ACE2 receptors expressed on the surface of pancreatic islet beta cells, leading to cytolysis and the emergence of “acute-onset type 1 diabetes”. Given that SARS-CoV-2 also uses the ACE2 receptor to enter cells, it has been hypothesized that it may trigger T1DM via a similar mechanism [20].

SARS-CoV-2 has been shown to cause severe diabetic ketoacidosis (DKA) at onset in individuals with newly diagnosed diabetes. However, at present, there is no conclusive evidence that SARS-CoV-2 alone causes type 1 diabetes [27].

A study by Bjerregaard-Andersen et al. [10] gathered data from 9 hospitals in Portugal to examine the impact of the COVID-19 pandemic on the incidence of type 1 diabetes in children and young adults. Data from 574 newly diagnosed T1DM cases (530 children and 44 adults) from 2017 to 2022 were analyzed. The results showed no significant increase in new T1DM cases during the COVID-19 pandemic. Moreover, significant differences were observed in neither blood glucose levels, HbA1c, or C-peptide at diagnosis, nor in the incidence of DKA compared to pre-pandemic years. These findings suggest that the COVID-19 pandemic is not associated with an increased incidence of T1DM in Portugal [10]. The absence of a significant increase in incidence in this study may be partly explained by confounding factors, such as delayed healthcare-seeking behavior and reduced access to diagnostic services during the early pandemic period. These challenges may have masked true incidence rates or shifted diagnoses outside the study window.

Genç et al. [29] published a case report of a 19-year-old female patient who developed COVID-19 and, one week later, presented with diabetic ketoacidosis (DKA). During hospitalization, she was also diagnosed with Hashimoto’s thyroiditis, highlighting SARS-CoV-2’s potential to trigger autoimmune responses. The interaction between SARS-CoV-2 and the renin–angiotensin–aldosterone system (RAAS) plays a key role in the pathophysiology of DKA and acute metabolic dysregulation. SARS-CoV-2 uses ACE2 as a functional receptor for cell entry, leading to a reduction in ACE2 expression due to internalization and degradation of the enzyme. The loss of ACE2 has several pathophysiological consequences: decreased ACE2 expression leads to accumulation of angiotensin II (Ang II), which negatively affects beta-cell function by inhibiting insulin secretion. At the same time, the direct entry of SARS-CoV-2 into islet cells exacerbates beta-cell damage, intensifying metabolic dysregulation and increasing the risk of hyperglycemia or new-onset diabetes [29].

In a multicenter study in Northwest London, Unsworth et al. [30] reported an increase in newly diagnosed T1DM cases in children aged up to 16 years, compared to previous years. High rates of DKA and severe clinical presentation were also observed, supporting the hypothesis of a possible association between SARS-CoV-2 and the onset of T1DM [30].

Overall, SARS-CoV-2 has emerged as a potential viral contributor to post-infectious diabetes, although the nature of this relationship remains complex and likely heterogeneous. It is important to distinguish between reports of type 1 and type 2 diabetes following COVID-19 infection. Many studies documenting new-onset diabetes during or after SARS-CoV-2 infection do not clearly differentiate between autoimmune (T1DM) and metabolic (T2DM) phenotypes, particularly in pediatric populations. For example, data from the N3C (National COVID Cohort Collaborative) registry indicated an increase in both types of diabetes post-infection, with significant heterogeneity by age, disease severity, and ethnicity [31].

In the context of T1DM, several immunopathogenic mechanisms have been proposed. SARS-CoV-2 can infect pancreatic islets via the ACE2 receptor, which is expressed on both beta and alpha cells, potentially leading to direct cytopathic effects or altered insulin/glucagon secretion [20]. The virus may also contribute to immune dysregulation, including epitope spreading—where the immune response to viral antigens expands to include self-antigens such as beta-cell proteins [32]. Additionally, failure of immune checkpoint control, particularly involving programmed cell death protein 1 (PD-1) and Cytotoxic T-Lymphocyte Antigen 4 (CTLA-4) pathways, may permit autoreactive T-cell activation and survival, compounding autoimmune risk [33].

Moreover, the disruption of the renin–angiotensin–aldosterone system (RAAS), especially through downregulation of ACE2, could exacerbate local inflammation and impair islet perfusion, promoting beta-cell stress or apoptosis [34]. These immunological and tissue-level events, particularly in genetically susceptible individuals, may collectively facilitate progression from islet autoimmunity to overt T1DM.

In contrast, the metabolic and inflammatory stress associated with severe COVID-19 may precipitate insulin resistance and beta-cell dysfunction consistent with T2DM, especially in individuals with predisposing factors such as obesity or prediabetes [35]. Distinguishing these outcomes remains a critical research priority as long-term follow-up data continue to emerge.

### 3.2. Molecular Mimicry: Coxsackie B Viruses, Rotavirus, Rubella, Influenza

Molecular mimicry is a key immunopathogenic mechanism in which viral antigens share structural or sequence similarity with pancreatic beta-cell autoantigens, leading to cross-reactive immune responses. This process may initiate or accelerate autoimmunity in genetically predisposed individuals without the need for direct beta-cell infection. One of the most well-documented examples involves Coxsackie B viruses (CVBs), in which a specific amino acid sequence—PEVKEK (Proline–Glutamic acid–Valine–Lysine–Glutamic acid–Lysine)—in the viral 2C protease closely resembles a region of glutamic acid decarboxylase 65 (GAD65), a major beta-cell autoantigen. This similarity can lead to the activation of autoreactive T cells and the development of islet autoantibodies in susceptible individuals [21,36]. Similarly, rotavirus has been implicated in molecular mimicry, particularly through its VP7 protein, which shares epitopes with GAD and insulinoma-associated antigen-2 (IA-2). Children with genetic predisposition have shown increased levels of islet autoantibodies following rotavirus infection, suggesting an autoimmune trigger via epitope similarity. The study by Honeyman et al. [3] examined the relationship between rotavirus infection and the development of islet autoantibodies in children at increased genetic risk for T1DM. A cohort of 360 children was followed from birth, and the appearance of autoantibodies (insulin autoantibodies-IAA, Glutamic Acid Decarboxylase-GAD65, and and insuline autoantibodies 2A-IA-2A) was recorded alongside the presence of anti-rotavirus antibodies. The findings suggest that rotavirus infection may activate or amplify the autoimmune process in beta cells via mechanisms of molecular mimicry, where viral antigens resemble pancreatic antigens. The study highlights the possible role of viral infections as environmental factors contributing to the onset of T1DM and underscores the need for further investigation into their role in disease pathogenesis [3].

Rubella virus, historically linked to congenital rubella syndrome and subsequent T1DM development, is also believed to act through mimicry-based mechanisms, although the exact antigenic targets remain undefined. Experimental evidence confirms that rubella virus can replicate in pancreatic islet cells, promoting inflammatory responses and potential autoimmune activation. However, despite this biological plausibility, epidemiologic evidence linking rubella infection to T1DM at the population level remains inconsistent. Rubella virus has historically been implicated in the development of type 1 diabetes mellitus (T1DM), particularly in the context of congenital rubella syndrome (CRS). Early epidemiological studies and case series demonstrated that children born with CRS had a significantly higher risk of developing T1DM, supported by findings of islet cell autoantibodies and compatible HLA profiles in these individuals. Experimental data also confirm that rubella virus is capable of replicating in pancreatic tissue, suggesting a mechanistic basis for beta-cell damage and subsequent autoimmune activation. However, more recent evaluations have questioned the epidemiological significance of rubella in T1DM pathogenesis. Despite the widespread adoption of rubella vaccination programs since the 1970s, no clear reduction in T1DM incidence has been observed in most countries. As discussed by Gale [37], this disconnect implies that rubella, while biologically plausible as a diabetogenic virus in cases of congenital infection, is unlikely to be a major contributor to T1DM in the general population. Furthermore, studies such as Christen et al. [7] have emphasized the limitations of serological data, which may be subject to cross-reactivity and do not reliably distinguish between past and recent infections. Another important consideration is the timing of infection. Rubella exposure in utero—especially during the first trimester—may have a greater impact on immune development than postnatal infection. This is supported by broader analyses of maternal viral exposures, including a systematic review and meta-analysis [38], which reported a modest but statistically significant association between maternal infection during pregnancy and increased T1DM risk in offspring. These findings highlight the role of early-life immune programming in modifying autoimmune susceptibility. In summary, while rubella virus retains mechanistic relevance—particularly in cases of CRS—the lack of population-level impact post-vaccination and limitations of existing serologic studies suggest its role in T1DM is likely limited and context-dependent.

Together, these examples highlight molecular mimicry as a plausible route through which viruses may contribute to beta-cell autoimmunity in T1DM

### 3.3. Persistent Infection and Immune Modulation

Beyond direct cytolysis and molecular mimicry, several viruses may contribute to the development of type 1 diabetes mellitus (T1DM) through persistent infection or immune modulation. In these cases, viruses do not immediately destroy beta cells but instead establish low-grade chronic infections or disrupt immune regulation, thereby sustaining inflammation and promoting autoimmunity. Coxsackie B viruses (CVBs) are a prime example, with evidence from the DiViD study showing viral RNA persistence in pancreatic islets of individuals newly diagnosed with T1DM. This persistent infection is accompanied by immune cell infiltration, suggesting that chronic viral presence may act as a continuous trigger for beta-cell autoimmunity [1,21]. Enterovirus-infected beta cells upregulate HLA class I and interferon-stimulated genes (ISGs), enhancing antigen presentation and inflammatory signaling [1,12]. Additionally, CVBs have been shown to impair natural killer (NK) cell function, reducing the expression of activating receptors (e.g., Natural Killer Group 2, member D- Natural Killer cell protein 46- NKG2D and NKp46) and increasing inhibitory signals, which weakens the immune system’s ability to eliminate infected cells and may facilitate immune dysregulation [39]. These host–virus dynamics suggest that chronic infection may promote autoimmunity even in the absence of acute cytolytic effects.

Cytomegalovirus (CMV) represents another example; while traditionally viewed as a bystander, some studies suggest that early-life CMV infection might modulate immune responses and delay the onset of T1DM, as shown in the DIPP study [19]. Cytomegalovirus (CMV) infection in early infancy has been investigated as a potential factor influencing autoimmunity development and progression to T1DM in genetically predisposed children. However, studies such as that by Aarnisalo et al. [18], which included 169 children with early autoantibody appearance and 791 healthy controls, did not find an association between CMV infection and the development of autoantibodies or progression to T1DM. In contrast, some recent data suggest that CMV infection may slow disease progression. Similar results were published in a study by Al-Hakami et al. [40]. These conflicting results may be partly due to methodological differences, especially in studies relying on serology, which can be limited by cross-reactivity, assay sensitivity, and the inability to determine the timing of infection. The timing of CMV exposure appears to be critical: prenatal or very early-life infection may shape immune responses differently than later exposure. Maternal transmission could influence fetal immune programming, although definitive evidence in T1DM is still lacking. Overall, the evidence does not support a significant role for CMV in T1DM pathogenesis; however, the presence of conflicting findings calls for further investigation [18]. Conversely, herpes simplex virus (HSV) and Epstein–Barr Virus (EBV) have been associated with increased risk of T1DM, particularly in younger individuals, although the mechanistic pathways remain incompletely understood [41]. These viruses may influence disease onset by altering cytokine profiles, modulating T-cell responses, or interacting with genetic risk loci.

The study by Wang et al. [41], involving a large population of 8179 patients with T1DM and 32,716 matched controls, revealed that infection with herpes simplex virus (HSV) is associated with an increased risk of developing T1DM, especially in individuals ≤18 years old. In contrast, infections with other human herpesviruses such as VZV (varicella-zoster), EBV, and HCMV showed no statistically significant association with T1DM risk. Additionally, comorbidities such as asthma, atopic dermatitis, Graves’ disease, Hashimoto’s thyroiditis, and enteroviral infection history were also associated with increased T1DM risk, whereas allergic rhinitis was not. These findings suggest HSV may play a role in T1DM pathogenesis, although existing literature remains limited [41]. In contrast, a study by Chen et al. [42], which investigated the association between herpes zoster (shingles) infection and the onset of T1DM using national registry data, found no statistically significant link between prior infection and increased T1DM risk. The findings suggest that the herpes zoster virus likely does not play a major role in T1DM pathogenesis [42].

In addition, both CMV and Epstein–Barr virus (EBV)—as persistent herpesviruses—are known to exert epigenetic effects on the host immune system. These include modulation of DNA methylation, histone modifications, and microRNA expression, which can influence the function of regulatory T cells, cytokine profiles, and antigen presentation pathways. Such changes may either promote or suppress autoimmunity, depending on the host’s genetic background and timing of viral exposure [43,44].

Influenza, particularly Influenza A (H1N1), has been associated with increased risk of type 1 diabetes mellitus (T1DM) through immune-mediated mechanisms rather than direct beta-cell infection. While there is limited evidence for molecular mimicry or direct cytolysis, influenza infection triggers a robust systemic immune response characterized by elevated levels of proinflammatory cytokines, such as IL-1β, IL-6, and TNF-α, which can contribute to bystander damage in pancreatic islets.

The prospective population-based study by Ruiz et al. [33] investigated the association between infection with pandemic influenza A (H1N1) and the subsequent risk of developing type 1 diabetes (T1DM) in a national sample of approximately 2.5 million individuals under the age of 30 in Norway. The authors tracked T1DM diagnoses from 2006 to 2014 and correlated these data with recorded influenza cases during the 2009–2010 pandemic. Although overall exposure to pandemic influenza was not associated with a statistically significant increase in the risk of T1DM (HR = 1.19, 95% CI: 0.97–1.46), patients with laboratory-confirmed H1N1 infection showed a significantly elevated relative risk (HR = 2.26, 95% CI: 1.51–3.38). These results support the hypothesis that severe respiratory viral infections may act as environmental triggers for the onset of T1DM in genetically predisposed individuals [33].

The study by Silvestri et al. [34] examined the presence of various viruses in children newly diagnosed with type 1 diabetes (T1DM), aiming to investigate potential viral triggers of disease activation. In a sample of 31 children, blood samples were analyzed for the presence of viruses such as Coxsackie A and B, echovirus, influenza viruses A/B, adenovirus, parainfluenza types 1–3, CMV, and RSV. The results confirmed the predominance of enteroviruses (Coxsackie and echovirus), but also recorded significant cases of parainfluenza virus infections, with seasonal peaks mainly in autumn and spring. Parainfluenza viruses were detected across various age groups and are thought to potentially play a role in initiating the autoimmune mechanism that leads to T1DM. These findings highlight the need for further investigation of non-enteroviral viruses as possible environmental triggers in the pathogenesis of the disease [34].

The study by Nishioka [35] explored the association between seasonal influenza and the incidence of new-onset type 1 diabetes (T1DM) in Japan. Data from national registries were analyzed over several years to assess whether flu outbreak periods correlated with an increase in T1DM onset. The results indicate a seasonal association, with increased incidence of newly diagnosed T1DM following influenza epidemics, supporting the hypothesis that viral infections such as influenza may act as activating factors in the pathogenesis of T1DM [35].

The evidence linking influenza infection to the development of type 1 diabetes mellitus (T1DM) remains moderate and inconclusive. Several epidemiological studies have reported temporal associations between seasonal influenza peaks and increased incidence of T1DM, particularly in children. However, it is essential to distinguish between studies based on laboratory-confirmed influenza (e.g., PCR or serological evidence) and those relying on clinical diagnosis codes, which may conflate true cases with other febrile illnesses. This distinction significantly impacts the reliability of findings, as studies based solely on clinical suspicion may overestimate associations. Additionally, the role of influenza vaccination should be considered when interpreting these trends. Variations in vaccine coverage and efficacy across populations and time periods may obscure potential correlations between natural influenza infection and T1DM onset. Some population-based studies have failed to find consistent patterns when vaccine uptake is factored into the analysis. Although influenza is primarily a respiratory virus, there is limited experimental evidence suggesting it may have indirect effects on the pancreas. In murine models, influenza infection has been associated with pancreatic inflammation and beta-cell stress, possibly via systemic cytokine responses. However, direct viral tropism for human pancreatic islets has not been conclusively demonstrated, and more research is needed to clarify whether influenza can act as a direct or indirect trigger in genetically susceptible individuals.

Varicella-zoster virus (VZV), the cause of chickenpox, has not been shown to directly infect pancreatic beta cells or to share epitopes with islet autoantigens. However, emerging evidence suggests it may play a modulatory role in the development of type 1 diabetes mellitus (T1DM). A study by Bougnères et al. [45] examined whether natural infection with the varicella-zoster virus (VZV) during early childhood influences the age of onset of type 1 diabetes (T1DM). The analysis was based on data from the French ISIS-DIAB cohort, which included 1604 children diagnosed with T1DM. The results showed that children who had contracted chickenpox before the age of two developed T1DM at a significantly older age (mean onset age: 8.7 years), compared to those with no history of infection (mean onset age: 7.7 years). These findings support the hypothesis that certain early viral infections may modulate the autoimmune response, affecting the course of T1DM onset. This finding positions varicella not as a classic diabetogenic virus but rather as a potential modulator of immune maturation, possibly influencing the tempo of autoimmune progression in genetically predisposed individuals [45]. This observation aligns with the **hygiene hypothesis**, which posits that reduced microbial exposure in early life may lead to dysregulated immune responses and increased risk of autoimmune diseases. Varicella, as a common early-life infection in the pre-vaccine era, may play a role in “educating” the immune system, promoting the development of **regulatory pathways** and **tolerance mechanisms** that delay or reduce the severity of autoimmune activation. One possible explanation is that varicella triggers regulatory T-cell (Treg) expansion or induces cytokine profiles (e.g., increased IL-10) that temporarily suppress autoreactive immune activity. Alternatively, early viral exposure might alter the gut microbiome or systemic immune tone in ways that confer protective immunomodulation. These findings support a broader view of virus–host interactions in T1DM as context-dependent, where timing, host immune status, and environmental factors determine whether a virus acts as a trigger or a modulator of disease.

## 4. Population Studies

The fact that the concordance rate of type 1 diabetes (T1DM) among identical twins is only 50% suggests that the disease’s pathogenesis depends on both genetic and environmental factors. The most significant genetic contribution comes from the HLA gene complex. At the same time, the seasonal variation in the onset of clinical T1DM—with fewer cases during the summer months in both the Northern and Southern Hemispheres—reinforces the hypothesis that environmental factors, possibly viral, play a role in the disease’s development.

Type 1 diabetes (T1DM) manifests after a long preclinical phase during which autoimmunity destroys the insulin-producing beta cells of the pancreas. This phase is marked by the presence of autoantibodies against insulin, glutamic acid decarboxylase (GAD), and the tyrosine phosphatase IA-2, as well as by the activation of T lymphocytes that target islet cells. Although the precise activation mechanisms remain unknown, the low concordance rate between monozygotic twins (50%) indicates that the disease’s pathogenesis relies on both genetic predisposition (primarily HLA) and environmental factors, likely viral in nature, as suggested by the seasonal pattern of T1DM onset [5,6]. The most important population studies are included in Table 2.

## 5. Mechanisms of Virus-Induced T1DM

The relationship between viral infections and type 1 diabetes (T1DM) is complex. Studies in mice have shown that some viruses may damage beta cells and trigger autoimmunity, while others may provide protection and have preventive benefits [8]. For example, viral data from non-obese diabetic (NOD) mice have shown that Coxsackie B3 virus (CB3) and lymphocytic choriomeningitis virus (LCMV) can protect against T1DM by promoting immune tolerance [27].

These include direct cytolysis of pancreatic beta cells, molecular mimicry, persistent infection, altered antigen presentation, and immune modulation via exosomal signaling. To aid conceptual clarity, a summary diagram has been added (Figure 1), mapping these mechanisms to representative viruses implicated in T1DM pathogenesis [27].

### 5.1. Direct Cytolysis

Direct cytolysis refers to the lytic destruction of pancreatic beta cells by viruses that can productively infect them. Several viruses, notably Coxsackie B viruses (CVBs), have demonstrated this capacity in both in vitro studies and animal models. CVBs can enter beta cells via the coxsackievirus and adenovirus receptor (CAR), leading to intracellular replication and subsequent cell lysis. This destruction results in the release of beta-cell autoantigens, amplifying the immune response and possibly initiating autoimmunity in genetically susceptible individuals. Histological studies, including those from the DiViD study, have detected viral RNA in the islets of patients with recent-onset T1DM, further supporting this mechanism [1,9]. Although lysis is an acute event, its immunological consequences may have lasting effects by providing inflammatory signals and antigenic material to local antigen-presenting cells.

### 5.2. Persistent Infection

Beyond acute cytolysis, certain viruses may establish persistent low-grade infections in pancreatic tissue, contributing to chronic immune activation and progressive beta-cell destruction. Persistent infections have been described particularly for enteroviruses, which can reside in islet cells without inducing rapid lysis. These infections may alter host cell function and modulate immune responses, for example, by upregulating MHC class I molecules on beta cells, increasing their visibility to cytotoxic T cells. Moreover, persistent viral presence can lead to ongoing innate immune stimulation, creating a pro-inflammatory microenvironment that undermines immune tolerance. Studies such as those from the DIPP and DiViD cohorts have found viral proteins or RNA persisting in the pancreas even after the initial infection subsides [13,47], suggesting a mechanism by which chronic low-level infection may sustain or amplify autoimmune progression.

### 5.3. Molecular Mimicry and Sequence Homology

Among the most studied indirect mechanisms is molecular mimicry, where viral peptides share significant sequence or structural similarity with self-antigens in pancreatic islet cells, for example, the following:

The PEVKEK motif in Coxsackievirus B4 VP1 protein mimics an epitope in GAD65, a major autoantigen in beta cells [48];Rotavirus VP7 has shown homology with IA-2, another islet cell antigen implicated in T1DM [49].

Autoantibodies targeting cytoplasmic components of pancreatic islets—such as glutamic acid decarboxylase (GAD), insulin, and tyrosine phosphatase proteins (IA-2/IA-2β)—can be detected months to years before the onset of hyperglycemia. The appearance of multiple autoantibodies especially signals a high likelihood of progression to symptomatic T1DM, often through a staged process, as follows: normoglycemic autoimmunity to dysglycemia and finally to the development of clinical disease [50]. Longitudinal cohort studies (e.g., DAISY and TEDDY) show that insulin autoantibodies (IAAs) typically arise in early childhood, followed by GAD autoantibodies (GADAs), with IA-2 and ZnT8 autoantibodies emerging later; the presence of ≥2 autoantibodies confers a >70% risk of developing diabetes within 10 years [51]. The gradual decline in pancreatic beta-cell function and secretory capacity suggests repeated autoimmune episodes contributing to progressive beta-cell loss.

### 5.4. Altered Antigen Presentation

Viral infections may also alter MHC class I and II presentation pathways in infected or nearby cells [52]. For instance, inflammatory signals induced by viruses can upregulate HLA class I molecules on beta cells, leading to enhanced presentation of beta-cell antigens and recruitment of autoreactive cytotoxic T lymphocytes [52]. Some viruses may also influence immunoproteasome processing, generating neoepitopes that bypass central tolerance [53].

### 5.5. Exosomes and Viral Antigen Transfer

Emerging evidence suggests that viruses can hijack exosomal pathways to disseminate their antigens or RNA without requiring direct cell lysis. These exosomes can be taken up by antigen-presenting cells, thereby contributing to bystander activation or immune priming at distal sites. Exosomal transfer may also enable antigen presentation in a context lacking co-stimulation, potentially favoring tolerance breakdown [54,55,56].

The review by Filippi and von Herrath [36] explores the dual role of viral infections in the pathogenesis of type 1 diabetes, presenting evidence supporting both a causal relationship and a potential protective effect. The authors analyze epidemiological and experimental findings on enteroviruses, especially Coxsackie B virus, which in some models can accelerate the onset of autoimmunity either through direct beta-cell infection or activation of the immune system. However, they also highlight the possibility that certain viruses—under specific conditions (such as timing of infection or viral load)—may exert immunoregulatory effects that prevent disease onset. Finally, the review emphasizes the importance of genetic background, age, and the microenvironment in modifying the impact of viruses on autoimmunity, underscoring the need for further research into the multifactorial nature of T1DM [1].

## 6. Discussion

Type 1 diabetes mellitus (T1DM) results from a multifaceted interplay between genetic predisposition and environmental exposures. Among the latter, viral infections have emerged as prominent contributors, with evidence from epidemiological, clinical, and experimental studies suggesting a role for several viral families in initiating or modulating beta-cell autoimmunity. This review examined the major direct and indirect mechanisms by which viruses can influence disease onset, including cytolytic beta-cell damage, molecular mimicry, persistent infection, altered antigen presentation, and exosomal immune modulation. Proposed mechanisms are depicted in Table 3.

Epidemiological data suggest an association between certain viral infections—such as mumps, rubella, and enteroviruses—and increased T1DM risk. However, conflicting evidence and the heterogeneity of study designs pose challenges to interpretation. For example, while enteroviral RNA has been detected in the pancreatic tissue of individuals with recent-onset T1DM, such findings are limited to a small number of postmortem studies and do not conclusively prove causality. It remains unclear whether viral presence precedes beta-cell destruction or represents a bystander effect. In addition, variation in detection techniques, timing of tissue sampling, and the absence of matched longitudinal controls complicate the picture further.

More recently, SARS-CoV-2 infection during the COVID-19 pandemic has been associated with increased reports of new-onset diabetes, although distinguishing T1DM from stress-induced hyperglycemia and ketosis-prone diabetes remains difficult in this context. Herpes simplex virus (HSV) has also been linked to elevated T1DM risk in younger individuals, whereas other herpesviruses, including cytomegalovirus (CMV), Epstein–Barr virus (EBV), and varicella-zoster virus (VZV), do not consistently demonstrate increased risk. Notably, some studies even suggest a protective or immunomodulatory effect of early exposure to certain viruses, possibly supporting the hygiene hypothesis.

Although molecular mimicry (e.g., PEVKEK in Coxsackie B4 vs. GAD65; rotavirus VP7 vs. IA-2) and persistent infection represent compelling models of immune activation, other mechanisms such as altered antigen presentation and immunoproteasome-mediated neoepitope generation are gaining attention. Viral-induced upregulation of MHC class I molecules on beta cells may enhance visibility to cytotoxic T lymphocytes, while exosomal transfer of viral RNA and antigens to antigen-presenting cells can stimulate immune responses even in the absence of overt infection.

While the collective body of evidence supports a link between viral infections and T1DM pathogenesis, it is important to acknowledge the methodological limitations of many cited studies. Sample sizes in some cohorts were small, limiting statistical power and generalizability. Additionally, heterogeneity in viral detection methods (e.g., PCR vs. serology) and specimen types (e.g., blood vs. stool) complicates direct comparison across studies. Several findings also stem from retrospective or observational designs, which are inherently vulnerable to bias and confounding. Future research with standardized methodologies, longitudinal sampling, and multicenter collaboration will be essential to validate and expand upon these findings.

Despite encouraging preclinical studies, there are no licensed antiviral- or vaccine-based interventions to prevent T1DM. However, promising developments are underway. Vaccines targeting enteroviruses—particularly Coxsackie B viruses—are being tested in early-phase clinical trials. Additionally, antiviral therapies aimed at controlling viral replication in genetically predisposed individuals are under investigation. Immunomodulatory approaches, including monoclonal antibodies such as teplizumab, have shown the capacity to delay disease progression in high-risk individuals. These findings reinforce the potential value of combining antiviral- and immune-based strategies to preserve beta-cell function.

Looking forward, several research priorities are critical to advancing our understanding of virus-induced autoimmunity. These include the following: (1) randomized trials of antiviral or vaccine strategies in high-risk pediatric populations; (2) longitudinal virome and seroconversion tracking in genetically susceptible individuals; (3) molecular and spatial profiling of virus-infected pancreatic tissue; and (4) improved animal models that integrate host genetics and virology. There is also an urgent need to standardize diagnostic criteria and outcome measures to enhance comparability across studies.

From a public health perspective, the identification of viral triggers in T1DM opens up the possibility of targeted prevention strategies. Pan-viral or virus-specific immunization of genetically at-risk groups, alongside surveillance and early detection of autoantibodies, could serve as a viable preventive framework. The integration of viral screening into newborn risk profiling and the development of early intervention protocols could, ultimately, reduce the incidence of this lifelong autoimmune condition.

## 7. Conclusions

In conclusion, while a direct causal link between viruses and T1DM remains unproven, converging lines of evidence support a contributory role, particularly among enteroviruses. Future studies should prioritize mechanistic clarity and translational potential, with the ultimate goal of moving from association to intervention.

## Figures and Tables

**Figure 1 biology-14-00981-f001:**
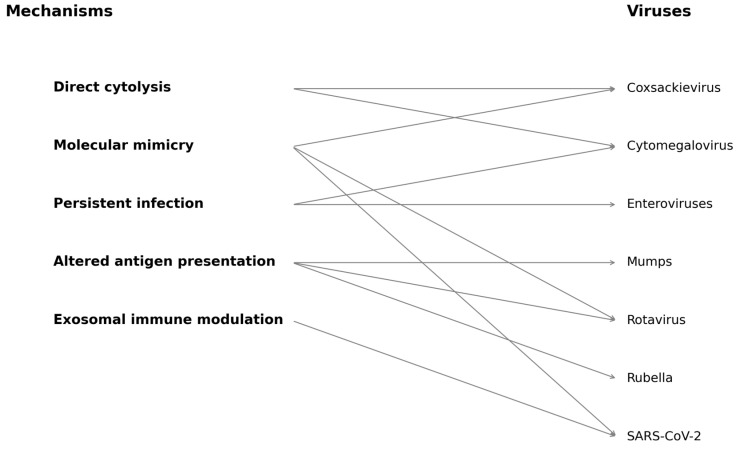
This diagram illustrates both direct and indirect pathways by which viral infections may contribute to pancreatic β-cell autoimmunity and destruction. Mechanisms include direct cytolysis, molecular mimicry, persistent infection, altered antigen presentation, and exosomal immune modulation. Specific viruses associated with each mechanism are indicated, including Coxsackievirus, rotavirus, cytomegalovirus, SARS-CoV-2, mumps, rubella, and enteroviruses. Arrows indicate proposed mechanistic associations based on the current literature.

**Table 1 biology-14-00981-t001:** Viruses implicated in the pathogenesis of type 1 diabetes mellitus.

Virus	Virus Family	Evidence Type	Mechanism of Action	Reference Studies
Coxsackie B (CVB)	Picornaviridae	Epidemiological, Histological, Animal	Beta-cell tropism, persistent infection, immune activation	[1,2,4]
Rotavirus	Reoviridae	Serological, Animal	Molecular mimicry, beta-cell damage	[3]
Mumps	Paramyxoviridae	Epidemiological, In vitro	Direct beta-cell infection, HLA I upregulation	[15,16]
Rubella	Togaviridae	Congenital Infection	Direct infection, molecular mimicry	[17]
CMV	Herpesviridae	Serological, Epidemiological	Possible protective modulation	[18,19]
SARS-CoV-2	Coronaviridae	Receptor/Pathophysiology	ACE * 2-mediated beta-cell entry, transient hyperglycemia	[10,20]

* ACE = angiotensin converting enzyme.

**Table 2 biology-14-00981-t002:** Population studies linking viral infections and T1DM onset (HR = hazard ratio).

Study/Cohort	Country	Virus	Sample Size	Study Design	Main Findings	Limitations	References
DAISY	USA	Enterovirus	~2500 children	Prospective, genetic-risk stratified	Enteroviral RNA linked to islet autoimmunity	Modest event rate; limited diversity	[4]
DiViD	Norway	Enterovirus	6 recent-onset adults	Pancreatic biopsy study	Viral RNA in pancreatic islets	Small sample size; no controls	[1]
TEDDY	EU/USA	Multiple viruses	>8000 children	Prospective, high-risk HLA	Temporal link between viruses and autoantibodies	Limited to genetically at-risk children	[2]
Finnish Registry	Finland	Rotavirus	>50,000 births	Retrospective cohort	Decline in T1DM after RV vaccine	Potential confounding; ecological analysis	[46]
N3C Database	USA	SARS-CoV-2	>1 million children	Retrospective EHR-based	Increased risk of new-onset diabetes post-COVID-19	Diabetes type often unspecified	[31]
DIPP	Finland	Enterovirus	>100,000 births	Prospective, genetically stratified	Strong temporal link between enterovirus and autoimmunity	Focused on high-risk genetic subgroups	[47]
ISIS-DIAB	France	Varicella	157 children with T1DM	Retrospective observational	Early varicella linked to delayed T1DM onset	Retrospective, limited sample	[45]
COVID-19 Portugal Study	Portugal	SARS-CoV-2	885 children (new T1DM cases)	Retrospective, national registry	Spike in T1DM incidence during/after COVID-19 waves	Unclear T1DM vs. T2DM classification; ecological design	[10]

**Table 3 biology-14-00981-t003:** Proposed mechanisms of virus-induced beta-bell destruction.

Mechanism	Description	Representative Viruses
Direct Cytolysis	Lytic destruction of beta cells via productive viral infection, releasing autoantigens and triggering immune responses.	Coxsackievirus B, Rubella
Persistent Infection	Low-grade, chronic infection in pancreatic tissue sustaining inflammation and antigen presentation.	Enteroviruses (e.g., Coxsackie B), CMV
Molecular Mimicry	Viral peptides share homology with beta-cell antigens, promoting cross-reactive autoimmune responses.	Coxsackievirus B4 (PEVKEK/GAD65), Rotavirus (VP7/IA-2)
Altered Antigen Presentation	Virus-induced inflammatory signals upregulate MHC molecules and immunoproteasome processing, revealing neoepitopes.	Enteroviruses, SARS-CoV-2
Exosomal Antigen Transfer	Viruses use exosomes to transfer RNA or protein to APCs, promoting immune priming without co-stimulation.	Enteroviruses, CMV, SARS-CoV-2

## Data Availability

Not applicable.

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
