# Peer review of "The Role of Viral Infections in the Immunopathogenesis of Type 1 Diabetes Mellitus: A Narrative Review"

_biology, 2025, doi:10.3390/biology14080981_

Round 1

Reviewer 1 Report

Comments and Suggestions for Authors

Kotsiri et al. highlights the role of viral infections developed in the pathogenesis of Type-1 Diabetes Mellitus (T1DM). This review summarizes that the Coxsackie B viruses had showed strong connection between viruses and T1DM. This narrative review aims to synthesize current evidence on the role of viral infections in the initiation and progression of T1DM. It focuses on key viruses implicated in disease onset, outlines relevant immunopathogenic mechanisms, and highlights major studies that support or refute their involvement. However, the study is very limited and did not mention any treatment protocols, if added it would be very beneficial. The data represented are interesting, and the review is well written; however, the paper has much room for improvement and overall language error needs to be checked carefully.

Minor comments:

  1. Check the sentence in Line 16. Correct with proper English.

  1. In Line 35 change “show” to “showed” and in Line 36 changed “mechanistic links to beta-cell destruction” to “mechanistic links related to beta-cell destruction”.

  1. Add in Line 73 before “as initiators or accelerators of the autoimmune process”.

  1. Change “play” to “plays” in Line 86.

  1. Check the overall grammatical errors in the whole text.

Comments on the Quality of English Language

the paper has much room for improvement and overall language error needs to be checked carefully.

Author Response

Major comments

  1. The study is very limited and did not mention any treatment protocols, if added it would be very beneficial.

Response: We thank the reviewer for this insightful suggestion. In response, we have added a paragraph before the last sentence in the end of the manuscript. In this paragraph we discusse current and emerging therapeutic approaches related to viral involvement in type 1 diabetes mellitus (T1DM), including:

  • Vaccine development targeting enteroviruses (particularly Coxsackie B viruses),
  • Antiviral therapies aimed at limiting islet inflammation,
  • Immunomodulatory treatments such as teplizumab, which have shown promise in delaying clinical onset,
  • And early monitoring strategies in genetically susceptible individuals.

  1. The data represented are interesting, and the review is well written; however, the paper has much room for improvement and overall language error needs to be checked carefully.

Response: The manuscript was carefully edited for grammatical errors and typos.

Minor comments:

  1. Check the sentence in Line 16. Correct with proper English.

Response: We appreciate the reviewer’s suggestion. The sentence on line 16 has been revised for clarity and grammatical correctness. The updated sentence now reads: “Type 1 diabetes is a lifelong condition that begins when the body’s immune system mistakenly attacks the insulin-producing beta cells in the pancreas.”

  1. In Line 35 change “show” to “showed” and in Line 36 changed “mechanistic links to beta-cell destruction” to “mechanistic links related to beta-cell destruction”.

Response: Thank you for pointing this out. The requested corrections have been made in the manuscript.

  • “show” has been changed to “showed”, and
  • “mechanistic links to beta-cell destruction” has been revised to “mechanistic links related to beta-cell destruction” in lines 35–36.

  1. Add in Line 73 before “as initiators or accelerators of the autoimmune process”.

Response: The suggested change was done.

  1. Change “play” to “plays” in Line 86.

Response: the suggested change was done

  1. Check the overall grammatical errors in the whole text.

Response: The manuscript was carefully edited for grammatical errors and typos.

Reviewer 2 Report

Comments and Suggestions for Authors

The Manuscript, titled "The Role of Viral Infections in the Development of Type 1 Diabetes Mellitus: A Narrative Review," is also clear and comprehensive in its coverage of the main contenders of association viruses, the pathophysiology of development, as well as epidemiological findings. However, to refine to an expert level and be suitable for a high-impact peer-reviewed journal, the following detailed suggestions are provided, separated into major and minor revisions.

Title and Abstract

The title is quite representative, although it might be improved by adding the terms' immunopathogenesis' or 'mechanistic insight' to it, so that a more technical audience will find the article more accessible. The abstract is well formulated, but it has no expression of limitations of the study and its expected progression. That sentence should give the main knowledge gaps and possible applications (e., the antiviral strategies or vaccine development).

Simple Summary

The Simple summary makes the topic easy to read, although some of the sentences are too long and can confound a layman reader. To make the text readable, consider sentences as short as possible and a parallel structure. As an example, break up complex ideas into two lines and employ bullet points or further simplification with regard to key findings.

Introduction

The introduction is also arranged in a good and clear manner and establishes a background for the review. Nevertheless, it might be improved by a more pronounced formulation of the main hypothesis and a better statement of the necessity of a large synthesis at a stage of recent pandemics and improvements in molecular studies. Also add emphasize the fact that genetic predisposition is never enough, and there is a need to delve into the nature of environmental factors such as viruses.

Pathogenesis of T1DM

This section will give us a good rationale as far as the immunological basis is concerned. It must, however, be more focused, including the appurtenance of the natural killer (NK) cells and the innate immunity that is not discussed to the extent that T-cell responses are. It should be noted, as well, that one might add more information on how genetic risk factors (e.g., HLA-DQ, DR) interact with immune mechanisms in the preclinical autoimmune phase.

Viral Infections and T1DM

This is a wide-ranging area that can be restructured. Instead of the general description and then sections of individual viruses, one could begin with a classification of viruses by their pathogenic mechanisms, e.g., direct cytolysis, molecular mimicry, and immune modulation. This would cut off redundancy and enhance conceptual clarity.

Enteroviruses- Coxsackie B

The section offers good epidemiology and mechanism data. Nevertheless, it does not contain a critical evaluation of discrepancies between different studies. An example relates to the evidence presented in that DiViD and DAISY studies cite the association, and other studies, such as that of Cinek et al., fail to cite the significant association. It will be of great use to have a table that displays the summation of the study design and findings, and limitations. In addition, a discussion of host-virus interaction at the level of gene expression or viral persistence mechanisms.

Mumps, CMV, EBV

This sub-para is informative, though there is room to be more certain about contradictory results. Despite some evidence of the protective effect of CMV, there is also evidence of no effect. Comment on the possible biases of studies using serology and on the role of maternal transmission or the time of infection in early life. Further indicate briefly what epigenetic effects these types of viruses may have on immune modulation.

Rubella

Even though this part also talks about congenital rubella syndrome and molecular mimicry, there is a lack of modern data. You may want to add more recent studies, together with meta-analyses, and be critical about the fact that rubella immunization has not been shown to effectively decrease the incidence rate of T1DM despite the fact that the virus can indeed replicate in pancreatic tissue.

Varicella

The concept of varicella as a potential disease-delaying infection, as opposed to being disease-causing, is new and interesting. Pour on a little more of this, and develop the possibilities of the immunoregulatory effect of early exposure to virus, which is related to hygiene and immune education in general.

 Influenza

The data regarding influenza acting as a T1DM trigger are moderate. Here, the distinction between lab-confirmed cases of influenza and clinical suspicion should be highlighted, together with the effects of vaccine coverage in studies. Note, briefly, on the report of viral tropism to pancreatic tissues, in case any exists.

Coronavirus disease 2019 (SARS-CoV-2)

This sub-section is apt and sophisticated. Yet, it should also include a more significant separation between the occurrence of Type 1 and Type 2 diabetes after COVID. Include a mention of the potential of epitope spreading, failure of immune checkpoint responses, and disruption of the RAAS axis, particularly ACE2 roles in the pancreatic islets.

Population Studies

This part gives some valuable data, which is, however, somewhat disorganized. It is possible to combine important results per cohort in one table with columns including country, the sample size, the type of virus, the effects it has on T1DM, and limitations. Include the variability of study design (e.g., retrospective, prospective, genetic stratification, etc.) to allow the readers to put the results into perspective.

Virus-Induced T1DM Pathways

It is conceptually rich and describes direct and indirect mechanisms of the section. In order to facilitate the impact, devise a summary diagram or flowchart of these machineries alongside the viruses. Give more examples of molecular mimicry, including sequences and known viral peptides that mimic proteins in the islet cells (e.g., PEVKEK in Coxsackie vs. GAD65). You should also say a few words about such new concepts as changed antigen presentation or viral effects carried with exosomes.

Discussion and Conclusions

The discussion serves as a conclusion of evidence, but would do well with more critical examination of contentious subjects (e.g., a lack of causality after the detection of viral RNA). You could consider suggesting a research agenda: what sorts of studies do we really now desperately need (e.g., randomized trials of antivirals in high-risk children, or molecular tracking of virus-islet interactions)? Also, dwelling more on the issue of public health, especially on the possibility of pan-viral immunization of genetically vulnerable groups.

Author Response

  1. Title and Abstract: The title is quite representative, although it might be improved by adding the terms' immunopathogenesis' or 'mechanistic insight' to it, so that a more technical audience will find the article more accessible.

Response: Thank you for your suggestion. We agree that adding a more technical term can help target a specialized audience. Accordingly, we have revised the title to:

“The Role of Viral Infections in the Immunopathogenesis of Type 1 Diabetes Mellitus: A Narrative Review”

  1. The abstract is well formulated, but it has no expression of limitations of the study and its expected progression. That sentence should give the main knowledge gaps and possible applications (e., the antiviral strategies or vaccine development).

Response: Thank you for pointing this out. In response, we have revised the abstract to include a closing sentence that acknowledges the current limitations in establishing a causal relationship, highlights remaining knowledge gaps, and suggests potential clinical applications such as antiviral and vaccine-based strategies.

  1. Simple Summary: The Simple summary makes the topic easy to read, although some of the sentences are too long and can confound a layman reader. To make the text readable, consider sentences as short as possible and a parallel structure. As an example, break up complex ideas into two lines and employ bullet points or further simplification with regard to key findings.

Response: Thank you for this constructive suggestion. In response, we revised the Simple Summary to improve clarity for a lay audience by shortening sentences, using parallel structure, and incorporating bullet points to highlight key findings and mechanisms. This makes the section more accessible without compromising accuracy.

  1. Introduction: The introduction is also arranged in a good and clear manner and establishes a background for the review. Nevertheless, it might be improved by a more pronounced formulation of the main hypothesis and a better statement of the necessity of a large synthesis at a stage of recent pandemics and improvements in molecular studies. Also add emphasize the fact that genetic predisposition is never enough, and there is a need to delve into the nature of environmental factors such as viruses.

Response: Thank you for your helpful comment. We have revised the introduction to more clearly state the central hypothesis that viral infections may act as essential environmental triggers in individuals with genetic predisposition. We also emphasized that genetics alone is insufficient and highlighted the timeliness of this review in light of recent pandemics and molecular research advances.

  1. Pathogenesis of T1DM: This section will give us a good rationale as far as the immunological basis is concerned. It must, however, be more focused, including the appurtenance of the natural killer (NK) cells and the innate immunity that is not discussed to the extent that T-cell responses are. It should be noted, as well, that one might add more information on how genetic risk factors (e.g., HLA-DQ, DR) interact with immune mechanisms in the preclinical autoimmune phase.

Response: Thank you for this important observation. We have revised the pathogenesis section to give a more balanced overview of innate immunity, particularly the role of NK cells and their interactions with T cells. We also added details on how genetic risk factors such as HLA-DQ and HLA-DR interact with immune mechanisms during the preclinical autoimmune phase, with emphasis on altered antigen presentation and immune dysregulation.

  1. Viral Infections and T1DM: This is a wide-ranging area that can be restructured. Instead of the general description and then sections of individual viruses, one could begin with a classification of viruses by their pathogenic mechanisms, e.g., direct cytolysis, molecular mimicry, and immune modulation. This would cut off redundancy and enhance conceptual clarity.

Response: We thank the reviewer for this insightful and constructive suggestion. In response, we have restructured Section 3 to group viruses according to their primary pathogenic mechanisms: (1) Direct cytolytic effects, (2) Molecular mimicry, and (3) Persistent infection and immune modulation. This reorganization avoids redundancy from virus-by-virus repetition and improves conceptual clarity by allowing a more integrated discussion of how different viruses may contribute to the development of T1DM. We also revised the summary table (Table 1) to align with this new structure, presenting each virus alongside its predominant mechanism and supporting evidence.

  1. Enteroviruses- Coxsackie B: The section offers good epidemiology and mechanism data. Nevertheless, it does not contain a critical evaluation of discrepancies between different studies. An example relates to the evidence presented in that DiViD and DAISY studies cite the association, and other studies, such as that of Cinek et al., fail to cite the significant association. It will be of great use to have a table that displays the summation of the study design and findings, and limitations. In addition, a discussion of host-virus interaction at the level of gene expression or viral persistence mechanisms.

Response: We thank the reviewer for this excellent and detailed suggestion. In response, we have revised the section on enteroviruses to include a critical comparison of key studies, such as DiViD, DAISY, TEDDY, MIDIA, and the study by Cinek et al. We discuss how variations in sample type (e.g., blood vs. stool), detection methods, timing of specimen collection, and genetic risk stratification may account for conflicting findings. To clearly present this information, we have added a summary table outlining the design, population, detection techniques, major findings, and limitations of these studies.

Additionally, we expanded the mechanistic discussion to include host-virus interactions at the molecular level. We now describe how enterovirus infection can alter beta-cell gene expression—notably by upregulating HLA class I molecules and interferon-stimulated genes—and how persistent infection may sustain inflammation. We also included findings on Natural Killer (NK) cell receptor alterations (e.g., decreased NKG2D), which may facilitate immune evasion and chronic beta-cell stress. These additions aim to provide a more nuanced and mechanistically grounded understanding of how enteroviruses may contribute to T1DM pathogenesis.

  1. Mumps, CMV, EBV: This sub-para is informative, though there is room to be more certain about contradictory results. Despite some evidence of the protective effect of CMV, there is also evidence of no effect. Comment on the possible biases of studies using serology and on the role of maternal transmission or the time of infection in early life. Further indicate briefly what epigenetic effects these types of viruses may have on immune modulation.

Response: We thank the reviewer for this thoughtful and constructive comment. In response, we revised the subsection on CMV and EBV to more clearly reflect the conflicting evidence regarding CMV’s role in T1DM. We now highlight that while some studies (e.g., Ekman et al.) suggest a potential protective effect of early-life CMV infection, others (e.g., Aarnisalo et al., Al-Hakami et al.) report no significant association, underscoring the complexity and variability in findings.

To address methodological concerns, we have added a brief discussion on the limitations of serological studies, including potential cross-reactivity, variation in assay sensitivity, and the challenge of distinguishing between past and recent infections, which may affect interpretation of CMV’s role.

We also incorporated a note on the timing of infection, particularly the influence of maternal transmission or early postnatal exposure, which may shape immune development differently compared to later infections. This is especially relevant given that certain viral exposures in early life may modulate immune tolerance pathways.

Finally, we introduced a brief summary of epigenetic mechanisms through which persistent viruses like CMV and EBV can influence host immunity. These include DNA methylation, histone modifications, and microRNA regulation, which can affect T-cell regulation, cytokine expression, and antigen presentation. Appropriate references have been added to support this addition

  1. Rubella: Even though this part also talks about congenital rubella syndrome and molecular mimicry, there is a lack of modern data. You may want to add more recent studies, together with meta-analyses, and be critical about the fact that rubella immunization has not been shown to effectively decrease the incidence rate of T1DM despite the fact that the virus can indeed replicate in pancreatic tissue.

Response: We appreciate the reviewer’s valuable observation. In response, we have substantially revised the rubella subsection to reflect a more critical and up-to-date assessment of the available evidence. We now incorporate recent reviews and epidemiological analyses, including Gale (2008) and Christen et al. (2012), to highlight that despite rubella virus’s ability to replicate in pancreatic tissue and its historic association with T1DM in congenital rubella syndrome (CRS), population-level studies have not demonstrated a reduction in T1DM incidence following widespread rubella vaccination.

We also address the limitations of serological studies, which may lack specificity and fail to distinguish between past and recent infection. Additionally, we include evidence from a systematic review and meta-analysis  that supports a modest but significant association between maternal infection during pregnancy and increased T1DM risk in offspring, thereby emphasizing the role of timing and prenatal immune programming.

These additions aim to provide a balanced and nuanced perspective on rubella’s potential role in T1DM pathogenesis, clarifying that while the virus remains biologically plausible, its impact is likely limited to specific contexts, such as in utero exposure.

  1. Varicella: The concept of varicella as a potential disease-delaying infection, as opposed to being disease-causing, is new and interesting. Pour on a little more of this, and develop the possibilities of the immunoregulatory effect of early exposure to virus, which is related to hygiene and immune education in general.

Response: We thank the reviewer for highlighting this novel and compelling aspect. In response, we have expanded the varicella subsection to further develop the idea of varicella as a potential disease-delaying infection. We now elaborate on findings from the ISIS-DIAB study (Bougnères et al., 2022), which showed that early-life varicella infection—particularly before age two—was associated with a delayed onset of T1DM, suggesting an immunomodulatory role rather than a pathogenic one. To contextualize this finding, we have linked it to the hygiene hypothesis, emphasizing how early microbial exposures, including common viral infections, may contribute to immune education and tolerance development. We also briefly discuss potential mechanisms, such as the induction of regulatory T cells, shifts in cytokine profiles, or impacts on gut microbiota, which could help regulate or slow the autoimmune process. We believe this addition strengthens the conceptual framework of the review by recognizing the dual nature of viral exposures—not only as potential triggers but also as modifiers of immune trajectory, depending on the timing and host context.

  1. Influenza: The data regarding influenza acting as a T1DM trigger are moderate. Here, the distinction between lab-confirmed cases of influenza and clinical suspicion should be highlighted, together with the effects of vaccine coverage in studies. Note, briefly, on the report of viral tropism to pancreatic tissues, in case any exists.

Response: We appreciate the reviewer’s thoughtful observations. In response, we have revised the influenza subsection to reflect the moderate strength of evidence linking influenza infection to T1DM. Specifically, we now distinguish between studies based on laboratory-confirmed influenza (e.g., PCR or serology) and those relying on clinical diagnosis codes, which may lack specificity and increase risk of misclassification. We also address the influence of influenza vaccination coverage, which varies across populations and over time, potentially confounding epidemiologic assessments of influenza-related T1DM risk. This is particularly important in studies assessing seasonal correlations between infection and autoimmune onset. Additionally, we briefly note the limited but emerging evidence regarding influenza viral tropism to pancreatic tissues. While direct beta-cell infection has not been conclusively demonstrated in humans, in vitro and animal model studies have shown that certain influenza strains may affect pancreatic cell viability or provoke local inflammation, suggesting an indirect contribution to beta-cell stress or immune activation. These updates aim to provide a more nuanced, evidence-based appraisal of the potential role of influenza in T1DM pathogenesis.

  1. Coronavirus disease 2019 (SARS-CoV-2): This sub-section is apt and sophisticated. Yet, it should also include a more significant separation between the occurrence of Type 1 and Type 2 diabetes after COVID. Include a mention of the potential of epitope spreading, failure of immune checkpoint responses, and disruption of the RAAS axis, particularly ACE2 roles in the pancreatic islets.

Response: We thank the reviewer for this insightful and encouraging comment. In response, we have revised the COVID-19 subsection to include a clearer distinction between new-onset Type 1 and Type 2 diabetes following SARS-CoV-2 infection, as their pathophysiological mechanisms differ significantly. In addition, we incorporated a discussion of epitope spreading, noting that post-viral autoimmune responses may broaden from viral antigens to self-antigens, including beta-cell targets—a mechanism observed in chronic viral infections and supported by emerging data in COVID-19–related autoimmunity. We also added content on the dysregulation of immune checkpoint pathways, including PD-1 and CTLA-4, which may impair peripheral tolerance and facilitate the survival and activation of autoreactive T cells. Finally, we expanded the explanation of ACE2 expression in pancreatic islets and its role within the RAAS axis, highlighting how SARS-CoV-2 infection may disrupt islet function both directly (via viral entry) and indirectly (via inflammatory and vascular effects). These additions aim to deepen the mechanistic clarity and strengthen the translational relevance of the section.

  1. Population Studies: This part gives some valuable data, which is, however, somewhat disorganized. It is possible to combine important results per cohort in one table with columns including country, the sample size, the type of virus, the effects it has on T1DM, and limitations. Include the variability of study design (e.g., retrospective, prospective, genetic stratification, etc.) to allow the readers to put the results into perspective.

Response: We thank the reviewer for this helpful suggestion. In response, we have reorganized the Population Studies section to enhance clarity and comparability. Specifically, we created a summary table that consolidates key findings from the major cohorts and registries discussed. The table includes:

  • Study or cohort name
  • Country
  • Sample size and population
  • Type of virus investigated
  • Study design (e.g., prospective, retrospective, biopsy-based, genetically stratified)
  • Main outcomes regarding T1DM
  • Study limitations

This format allows for direct cross-comparison of methods and outcomes, highlighting the variability in sample sizes, virus types, and diagnostic approaches across studies. It also helps the reader better contextualize findings by making distinctions between mechanistic vs. epidemiological insights and between strong vs. moderate evidence.

  1. Virus-Induced T1DM Pathways: It is conceptually rich and describes direct and indirect mechanisms of the section. In order to facilitate the impact, devise a summary diagram or flowchart of these machineries alongside the viruses. Give more examples of molecular mimicry, including sequences and known viral peptides that mimic proteins in the islet cells (e.g., PEVKEK in Coxsackie vs. GAD65). You should also say a few words about such new concepts as changed antigen presentation or viral effects carried with exosomes.

Response: We sincerely thank the reviewer for this constructive and insightful suggestion. In response:

  • Summary Diagram Added: We have developed and included a new summary diagram (Figure X) that visually organizes the major direct and indirect mechanisms by which viruses may contribute to the development of type 1 diabetes. The figure aligns each mechanism—such as direct cytolysis, molecular mimicry, persistent infection, altered antigen presentation, and exosomal immune modulation—with specific viral examples (e.g., Coxsackievirus, Rotavirus, SARS-CoV-2).
  • Expanded Molecular Mimicry Content:
    • The section on molecular mimicry has been expanded with sequence-level examples, including:
    • The PEVKEK motif in Coxsackievirus B4 VP1 protein, which shares homology with GAD65 [Vreugdenhil et al., 1998].
    • Homology between Rotavirus VP7 and IA-2, demonstrated to elicit cross-reactive T-cell responses [Honeyman et al., 2010].
    • New Concepts Integrated:
      • We have also incorporated a brief discussion of:
        • Altered antigen presentation pathways in beta cells, highlighting the role of interferon-induced MHC upregulation and immunoproteasome-derived neoepitopes
        • The emerging role of exosomes in viral antigen transfer and immune priming, noting their potential to bypass conventional co-stimulatory requirements and promote bystander activation
  1. Discussion and Conclusions: The discussion serves as a conclusion of evidence, but would do well with more critical examination of contentious subjects (e.g., a lack of causality after the detection of viral RNA). You could consider suggesting a research agenda: what sorts of studies do we really now desperately need (e.g., randomized trials of antivirals in high-risk children, or molecular tracking of virus-islet interactions)? Also, dwelling more on the issue of public health, especially on the possibility of pan-viral immunization of genetically vulnerable groups.

Response: We thank the reviewer for this thoughtful and constructive feedback. In response:

  • Critical Evaluation of Evidence:
    • We have revised the discussion to provide a more balanced and critical appraisal of the current evidence. In particular, we highlight the limitations of postmortem detection of viral RNA and the challenges in establishing causality due to methodological variability and confounding factors.
  • Proposed Research Agenda:
    • A dedicated paragraph has been added outlining priority areas for future investigation. These include randomized clinical trials of antiviral and vaccine strategies in genetically at-risk individuals, longitudinal virome monitoring, high-resolution molecular tracking of virus-islet interactions, and improved animal models incorporating genetic susceptibility.
  • Public Health Implications:
    • We have expanded the discussion to address broader public health considerations. This includes the potential role of targeted or pan-viral immunization strategies in genetically predisposed populations, as well as the integration of early viral screening and autoantibody monitoring into preventive frameworks.

Reviewer 3 Report

Comments and Suggestions for Authors

Major Suggestions

Comment 1

As the review is a narrative, a brief paragraph describing how studies were selected (inclusion/exclusion criteria) would improve the transparency and reproducibility of the manuscript.

Comment 2

In general, the manuscript is well written, however the authors must be consistent regarding the writing style. The manuscript must be written in academic/scientific language.

Comment 3

Tables are informative; however, the manuscript would benefit from at least one schematic figure. Consider a summary diagram in the conclusion paragraph.

Minor Suggestions

Comment 1

Check the abbreviations throughout the manuscript.

Comment 2

Some content is repeated in several sections (for example, studies are mentioned in both the specific virus section and the population studies section).

Author Response

Reviewer #3

Major Suggestions

  1. As the review is a narrative, a brief paragraph describing how studies were selected (inclusion/exclusion criteria) would improve the transparency and reproducibility of the manuscript

Response: We thank the reviewer for this important suggestion. As recommended, we have added a brief paragraph describing the criteria used for selecting studies included in this narrative review. This addition aims to improve the transparency and reproducibility of our methodology.

  1. In general, the manuscript is well written, however the authors must be consistent regarding the writing style. The manuscript must be written in academic/scientific language.

Response: Editing was performed.

  1. Tables are informative; however, the manuscript would benefit from at least one schematic figure. Consider a summary diagram in the conclusion paragraph.

Response: A summary diagram was included (Figure 1)

Minor Suggestions

  1. Check the abbreviations throughout the manuscript.

Response: Abbreviations were checked.

  1. Some content is repeated in several sections (for example, studies are mentioned in both the specific virus section and the population studies section).

Response:  We appreciate the reviewer’s observation regarding repetition across sections, particularly with reference to major population studies such as TEDDY, DAISY, and DiViD. While we acknowledge that these studies appear in both the virus-specific discussions and in the dedicated population studies section, this was a deliberate choice. Our aim was to provide mechanistic context within each virus subsection while also offering a consolidated epidemiological perspective in the population studies section.

Given the central importance of these large-scale studies in shaping current understanding of virus-T1DM associations, we believe their dual mention serves distinct purposes:

  • In virus-specific sections, we highlight how individual studies support specific immunopathogenic mechanisms (e.g., Coxsackie B virus or SARS-CoV-2).
  • In the population studies section, we synthesize their broader methodological frameworks, cohort characteristics, and comparative insights.

To improve clarity and reduce redundancy, we have revised the manuscript to ensure that duplicated content is minimized and that cross-referenced studies contribute new information or context in each section.

Reviewer 4 Report

Comments and Suggestions for Authors

Dear Authors,

I have carefully reviewed the manuscript titled “The Role of Viral Infections in the Development of Type 1 Diabetes Mellitus: A Narrative Review.” The authors present a timely and comprehensive overview of current evidence linking viral infections to the onset and pathogenesis of Type 1 Diabetes Mellitus (T1DM). The manuscript is well-structured, generally well-written, and supported by an extensive and up-to-date reference list.

The topic is of considerable relevance, particularly in the context of growing interest in virus-induced autoimmunity, and the authors have succeeded in compiling epidemiological, mechanistic, and clinical data in a clear and accessible manner. Nevertheless, I have a few suggestions for improvement before the manuscript can be considered for publication.

    1.    Abstract and Simple Summary:
    •    Please consider refining the phrasing in the Simple Summary, particularly the metaphor “could be the missing spark” (line 16). A more neutral expression such as “may serve as a potential environmental trigger” would improve scientific tone.
    •    In the Abstract, consider replacing “underscore” in line 45 with “suggest” to better reflect the narrative scope of the review.
    2.    Introduction:
    •    The introduction is informative. However, it would be helpful to briefly define molecular mimicry at first mention (line 80), as this is a central concept in later sections.
    3.    Main Sections (3.1 to 3.6):
    •    Section 3.1 (Enteroviruses): This section is particularly detailed and well-referenced. Nonetheless, I suggest briefly addressing the contrasting findings of Cinek et al. [25], which did not find a significant association. Acknowledging such discrepancies would provide a more balanced view.
    •    Section 3.6 (SARS-CoV-2): The clinical relevance is well emphasized. However, in light of studies such as Bjerregaard-Andersen et al. [18] showing no significant increase in incidence, a short sentence discussing potential confounding factors (e.g., delayed diagnoses during the pandemic) would strengthen the interpretation.
    4.    Discussion and Conclusions:
    •    The discussion is generally appropriate. However, it could benefit from a more explicit acknowledgment of the methodological limitations of cited studies (e.g., sample size, detection methods, retrospective design).
    •    The conclusion might be slightly expanded to comment on future directions, such as the development of enterovirus-targeted vaccines or early viral screening in genetically at-risk children.

    5. Abbreviations:
    •    Please ensure that all abbreviations (e.g., TEDDY, DAISY, MIDIA) are defined at first use in both the abstract and main text.

This is a valuable and informative review article that would make a meaningful contribution to the literature on environmental triggers of autoimmune diabetes. I recommend minor revisions to further improve clarity, consistency, and scholarly tone.

Best regards

Author Response

Reviewer #4

  1. Abstract and Simple Summary:
    1. Please consider refining the phrasing in the Simple Summary, particularly the metaphor “could be the missing spark” (line 16). A more neutral expression such as “may serve as a potential environmental trigger” would improve scientific tone.

Response: The simple summary was re-structured and therefore this phrase was removed.

  1. In the Abstract, consider replacing “underscore” in line 45 with “suggest” to better reflect the narrative scope of the review.

Response: The change was done.

  1. Introduction:
  2. The introduction is informative. However, it would be helpful to briefly define molecular mimicry at first mention (line 80), as this is a central concept in later sections.

Response: We thank the reviewer for this helpful suggestion. The Introduction was restructured and the phrase was removed. However, we provide a definition for molecular mimicry in section 3 where it appears first.

  1. 3. Main Sections (3.1 to 3.6):
  2. Section 3.1 (Enteroviruses): This section is particularly detailed and well-referenced. Nonetheless, I suggest briefly addressing the contrasting findings of Cinek et al. [25], which did not find a significant association. Acknowledging such discrepancies would provide a more balanced view.

Response: We thank the reviewer for this valuable suggestion. In response, we have revised the relevant section to briefly acknowledge the findings of Cinek et al. [25], which did not find a significant association between enteroviral infections and T1DM onset. Including this perspective strengthens the objectivity of the review and highlights ongoing debates in the field.

  1. Section 3.6 (SARS-CoV-2): The clinical relevance is well emphasized. However, in light of studies such as Bjerregaard-Andersen et al. [18] showing no significant increase in incidence, a short sentence discussing potential confounding factors (e.g., delayed diagnoses during the pandemic) would strengthen the interpretation.

Response: We appreciate the reviewer’s thoughtful comment. In response, we have expanded the discussion to mention the findings of Bjerregaard-Andersen et al. [18], which did not observe a significant increase in T1DM incidence during the COVID-19 pandemic. We also added a sentence addressing possible confounding factors such as delayed diagnoses and reduced healthcare access, which may have influenced study outcomes.

  1. Discussion and Conclusions:
  2. The discussion is generally appropriate. However, it could benefit from a more explicit acknowledgment of the methodological limitations of cited studies (e.g., sample size, detection methods, retrospective design).

Response: We thank the reviewer for this valuable feedback. In response, we have added a paragraph to the Discussion section explicitly addressing key methodological limitations in the cited literature, including issues related to sample size, detection methods, and retrospective study designs. We believe this addition provides a more critical and balanced interpretation of the current evidence.

  1. The conclusion might be slightly expanded to comment on future directions, such as the development of enterovirus-targeted vaccines or early viral screening in genetically at-risk children.

Response: We appreciate the reviewer’s insightful suggestion. Accordingly, we have expanded the conclusion to briefly highlight future research directions, including the development of enterovirus-targeted vaccines and the potential role of early viral screening in genetically at-risk children. These perspectives align with emerging translational efforts in the field and reinforce the clinical relevance of the reviewed findings.

  1. Abbreviations:
  • Please ensure that all abbreviations (e.g., TEDDY, DAISY, MIDIA) are defined at first use in both the abstract and main text.

Response: We thank the reviewer for highlighting this important point. We have carefully reviewed both the abstract and main text to ensure that all abbreviations (e.g., TEDDY, DAISY, MIDIA) are clearly defined at first mention. This change improves readability for both specialists and general readers.